# Adaptation of health systems to climate change-related infectious disease outbreaks in the ASEAN: Protocol for a scoping review of national and regional policies

Adriana Viola Miranda [1]*, Bony Wiem Lestari[1,2,3], Annisa Indrarini[3], Fadilah F. Arsy[3], Saut Sagala[3,4], Mizan Bustanul Fuadi Bisri[3,5], Don Eliseo Lucero-Prisno, III[6,7,8]

1 Research Center for Care and Control of Infectious Diseases (RC3ID), Universitas Padjadjaran, Bandung, Indonesia, 2 Department of Public Health, Universitas Padjadjaran, Bandung, Indonesia, 3 Resilience Development Initiative, Bandung, Indonesia, 4 School of Architecture, Planning and Policy Development, Bandung Institute of Technology (ITB), Bandung, Indonesia, 5 Kobe University, Kobe, Japan, 6 Department of Global Health and Development, London School of Hygiene and Tropical Medicine, London, United Kingdom, 7 Faculty of Management and Development Studies, University of the Philippines Open University, Los Baños, Laguna, Philippines, 8 Faculty of Public Health, Mahidol University, Bangkok, Thailand

* adriana.viola@alumni.ui.ac.id

**Data Availability Statement:** No datasets were generated or analysed during the current study. All

## Abstract

### Background

The Association of South-East Asian Nations (ASEAN) member states (AMS) are among the countries most at risk to the impacts of climate change on health and outbreaks being a major hotspot of emerging infectious diseases.

### Objective

To map the current policies and programs on the climate change adaptation in the ASEAN health systems, with particular focus on policies related to infectious diseases control.

### Methods

This is a scoping review following the Joanna Briggs Institute (JBI) methodology. Literature search will be conducted on the ASEAN Secretariat website, government websites, Google, and six research databases (PubMed, ScienceDirect, Web of Science, Embase, World Health Organization (WHO) Institutional Repository Information Sharing (IRIS), and Google Scholar). The article screening will be based on specified inclusion and exclusion criteria. Policy analysis will be conducted in accordance with the WHO operational framework on climate-resilient health systems. Findings will be analyzed in the form of narrative report. The reporting of this scoping review follows the Preferred Reporting Items for Systematic Reviews and Meta-Analyses extension for Scoping Reviews (PRISMA-ScR).

relevant data from this study will be made available upon study completion.

**Funding:** The author(s) received no specific funding for this work.

**Competing interests:** The authors have declared that no competing interests exist.

## Ethics and dissemination

Ethical approval is not required for this study as this is a scoping review protocol. Findings from this study will be disseminated through electronic channels.

## Introduction

According to the World Health Organization (WHO), climate change is the biggest health threat facing humanity. Climate change disrupts health systems globally as extreme weather events lead and contribute to infectious disease outbreaks, food system disruptions, and other climate-related health problems. Several factors are associated with the increase in infectious disease outbreaks: global warming is known to allow more favourable conditions for transmission of vector-borne diseases, such as dengue; food system disruptions lead to more food-borne diseases; and disasters such as flooding cause the transmission of water-borne diseases [1]. As the climate change progresses, climate-related health disruptions will be more evident. Warming of oceans results into more precipitation and heavy rainfalls causing more floods. Recognizing these issues, countries globally have pledged to prioritize climate mitigation, resilience and adaptation through national, regional, and global cooperation frameworks [1, 2].

The member states of the Association of South-East Asian Nations (ASEAN) are among the countries most at risk with the health impacts of climate change. Even without climate change, the ASEAN countries face huge challenges on outbreak control: it is a major hotspot for emerging infectious diseases due to its geographical location, proximity to the Pacific Ocean, migration rate, deforestation, and increase in food production [3]. Vector-borne outbreaks such as dengue are relatively more prevalent in this region of the world compared to other countries, with reported cases ranging between approximately 3,000 (Cambodia) to 145,000 (Vietnam) in 2022 [4]. A 2015 report from the WHO indicates that in both high and low emission scenarios, climate change is projected to increase dengue and malaria risks in ASEAN countries as it allows more favourable conditions for transmission (S1 Fig) [5–11].

To strengthen climate adaptation across sectors, governments in the 2010 United Nations Climate Change Conference (more commonly known as the Conference of the Parties/COP) established the National Adaptation Plan (NAP) process. Countries are to develop medium & long-term plans on climate mitigation and adaption [12]. All ASEAN member states have started the NAP process with the United Nations Framework Convention on Climate Change (UNFCCC), however, as of 26 September 2022, only Cambodia has submitted their NAP. Within the ASEAN, climate change is included as one of the goals of the ASEAN Health Cluster 2 on Responding to All Hazards and Emerging Threats (as outlined by the ASEAN Post-2015 Health Development Agenda 2021–2025) [13]. Several cooperative frameworks have been established to address this goal. However, no study has been conducted so far that assesses how AMS and the ASEAN as a regional body adopted climate change on their policies including the adaptation plans. Therefore, in this study, we aimed to map the evidence of climate-related health adaptation policies and programs in the ASEAN, both at the country level and within the ASEAN as a regional body, with particular focus on policies related to infectious diseases control.

## Methods

This is a scoping review aimed at assessing the current landscape of climate-related health policies among AMS and the ASEAN that follows the Joanna Briggs Institute (JBI) methodology.

The policies included are analyzed based on the WHO operational framework for building climate-resilient health systems [14]. These combined methods of scoping review and framework analysis follow prior studies by Ye et al (2021) and Adom et al (2021) [15, 16]. The reporting of this scoping review follows the Preferred Reporting Items for Systematic Reviews and Meta-Analyses extension for Scoping Reviews (PRISMA-ScR) (S1 Checklist).

### Identifying research questions

How climate change adaptation are included into health policies and actions in the ASEAN health systems, with particular focus on infectious diseases control?
 Sub-questions:

1. What are the characteristics of existing policies and actions for climate change adaptation in health systems among ASEAN member states and the ASEAN region, with particular focus on infectious diseases control?

2. To what extent do the current climate-related health policy documents from AMS adopt the WHO Operational Framework for Climate-Resilient Health Systems?

3. What are the indicators that have been used to assess for climate change adaptation in health systems among ASEAN member states and the ASEAN region, with particular focus on infectious diseases control?

### Eligibility of the research questions

The eligibility of the primary research question was assessed using the Population, Concept, Context (PCC) framework [17].

- P-Population: Humans.

- C-Concept: Policies and actions on climate change adaptation in health systems, focusing on infectious disease outbreaks.

- C-Context: ASEAN member states (AMS) and the ASEAN.

### Search strategy

We will conduct a systematic search on the ASEAN Secretariat website, government websites, Google, and six research databases (PubMed, ScienceDirect, Web of Science, Embase, WHO Institutional Repository Information Sharing (IRIS), and Google Scholar) to identify relevant policy documents and grey literature. The search will be conducted using a combination of keywords (and/or MeSH terms, if applicable): "Climate change", "healthcare system", "health preparedness", "infectious diseases", "Southeast Asia", "ASEAN", and the names of each ASEAN country (Brunei Darussalam, Cambodia, Indonesia, Laos, Malaysia, Myanmar, Philippines, Singapore, Thailand, Vietnam). S1 Appendix shows the keywords that will be used for each database. Duplicated documents from the research databases will be filtered using Zotero application.

### Study selection and eligibility criteria

The inclusion criteria of this scoping review are:

1. Policy documents and studies on climate change adaptation policies and actions, with particular focus on infectious diseases, in ASEAN health systems.

2. The documents should be published within 20 years (between January 2003 to January 2023) to allow a comprehensive analysis of the current climate change adaptation policies, particularly on infectious diseases, in ASEAN. If there are two or more policies published by a country or in ASEAN, we will only include policies that are still in effect.

No language restrictions will be imposed. Translations of original articles will only be conducted if the article is relevant. We will exclude conference proceedings, books, blogs, news articles, and articles that are not available in full-text form. Two reviewers, AVM and BWL, will independently assess the eligibility and inclusion of the studies by screening the titles and abstracts, followed by full-text assessments. Any differences will be resolved through consensus.

## Data extraction, summary and reporting

To summarize our search results, we will use the PRISMA flow chart. Two reviewers will extract data from the included documents into a matrix containing information about the country, name of documents, type of documents, issuing body, and type of addressed infectious diseases-related issues (vector-borne, water-borne, and/or food-borne diseases, and their effects to health systems). We will further look into the concordance of these policies to the WHO's ten key components for building climate-resilient health systems [14] and on whether the policies included key performance indicators to assess the climate change adaptation process among AMS health systems (Table 1). We will also collect information on whether the AMS has any policy change in the past 20 years. This data will be the basis for a summary narrative discussing the current national and regional policies on the topic, the infectious diseases that are the focus of these policies, and the remaining gaps that need to be addressed.

## Analytical framework

To analyze the included policies, we will use the framework obtained from the WHO's ten key components for building climate-resilient health systems (Table 1): leadership and governance; health workforce; vulnerability, capacity & adaptation assessment; integrated risk monitoring & early warning; health & climate research; climate resilient and sustainable technologies & infrastructure; management of environmental determinants of health; climate-informed health programs; emergency preparedness and management; climate and health financing [14].

## Ethics and dissemination

This study does not require ethical approval as it is a scoping review protocol. The findings will be published in scientific journals and disseminated electronically. All relevant literature search results such as government documents, journal articles, and publications, will be made available in a public repository.

## Strengths and limitations of this study

This is the first study that systematically reviews the climate-related health policies and collaborative frameworks in the ASEAN. The findings of this scoping review will provide comprehensive information on climate-related health policies and programs for AMS and ASEAN policymakers. The scoping review is guided by the Joanna Briggs Institute (JBI) methodology

**Table 1. Extracted data from national policy documents.**

| Extracted Data | | Description |
|---|---|---|
| Country | | ASEAN countries (Brunei Darussalam, Cambodia, Indonesia, Laos, Malaysia, Myanmar, Philippines, Singapore, Thailand, Vietnam), or ASEAN subregion and region |
| Type of document | | Policy document, regional announcement |
| Title of document | | The title of the policy documents of each country |
| Issuing body | | The governmental body/organization publishing the policy documents |
| Type of addressed infectious diseases-related issues | | The type(s) of diseases (vector-borne, water-borne, food-borne diseases) and infectious diseases-related health systems issues addressed by the policies (e.g. the need for disease-specific capacity building, resilient infrastructure against flood and other weather events) |
| Concordance to WHO's ten key components for building climate resilience | 1) Leadership and governance | Cross-sectoral leadership and strategic planning, particularly in health-related sectors e.g. water and sanitation, food, energy and urban planning |
| | 2) Health workforce | Strengthening of technical and professional capacity of healthcare workers (e.g. through additional professional training), organizational capacity, and institutional collaborative capacity in responding effectively to climate-related health risks |
| | 3) Vulnerability, capacity & adaptation assessment | Assessment of vulnerable populations and health systems capacity that inform climate adaptation interventions, which can include vulnerability and risk mapping, modelling, scenario development, as well as assessment of health system capacity and performance, economic impact, health impact, and specific hazards |
| | 4) Integrated risk monitoring & early warning | The use of epidemiological surveillance and early detection tools to monitor environmental determinants of health and inform early emergency warning systems, including timely warnings to decision-makers and stakeholders |
| | 5) Health & climate research | Support for multidisciplinary national research agenda and dissemination to policymakers, e.g. access to research data, partnerships, training and financing |
| | 6) Climate resilient and sustainable technologies & infrastructure | The use of new technologies for better healthcare delivery to enhancing climate resilience and ensure sustainability of health operations despite climate-related risks |
| | 7) Management of environmental determinants of health | Multisectoral public health prevention programs to control environmental determinants of health, e.g. air quality, water quantity & quality, food security, housing, and waste management |
| | 8) Climate-informed health programs | Climate-informed health programming and operations that take into account current and projected future climate variability |
| | 9) Emergency preparedness and management | Climate-informed emergency preparedness plans, systems and management, including community empowerments |
| | 10) Climate and health financing | Health-specific climate financing and for sectors influencing health |
| Key performance indicators | | Health-specific key performance indicators mentioned in the policies (if any) |

for scoping review, ensuring clear and replicable literature search process. In addition, our policy analysis is guided by the WHO operational framework on climate-resilient health systems, allowing for future comparative analysis. The limitations of this study are the English keywords used for the literature search, reducing the possibility of including articles in other languages, and the year criteria (20 years), although this is intended to give focus to current policies.

## Supporting information

**S1 Fig. Dengue fever and malaria transmission in ASEAN countries, assumming high emission scenario.**
(DOCX)

**S1 Checklist. Preferred Reporting Items for Systematic Reviews and Meta-Analyses extension for Scoping Reviews (PRISMA-ScR) checklist.**
(DOCX)

**S1 Appendix. Database search strategy.**
(DOCX)

## Author Contributions

**Conceptualization:** Adriana Viola Miranda, Bony Wiem Lestari.

**Supervision:** Bony Wiem Lestari, Don Eliseo Lucero-Prisno, III.

**Visualization:** Adriana Viola Miranda.

**Writing – original draft:** Adriana Viola Miranda.

**Writing – review & editing:** Adriana Viola Miranda, Bony Wiem Lestari, Annisa Indrarini, Fadilah F. Arsy, Saut Sagala, Mizan Bustanul Fuadi Bisri, Don Eliseo Lucero-Prisno, III.

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
