## [Decision Letter · Decision Letter 0]

29 Mar 2023

PONE-D-23-02161Adaptation of health systems to climate change in the ASEAN: Protocol for a scoping review of national and regional policiesPLOS ONE

Dear Dr. Miranda,

Thank you for submitting your manuscript to PLOS ONE. After careful consideration, we feel that it has merit but does not fully meet PLOS ONE’s publication criteria as it currently stands. Therefore, we invite you to submit a revised version of the manuscript that addresses the points raised during the review process.

We look forward to receiving your revised manuscript.

Kind regards,

Seo Ah Hong, PhD

Academic Editor

PLOS ONE

2. Please submit the filled PRISMA- ScR checklist

3. Please upload a copy of your study protocol that was approved by your ethics committee/IRB as a Supporting Information file. By the study protocol, we mean the complete and detailed plan for the conduct and analysis of the trial approved by the ethics committee/IRB. Please send this in the original language. If this is in a language other than English, please also provide a translation. [https://journals.plos.org/plosone/s/submission-guidelines#loc-guidelines-for-specific-study-types]

Reviewers' comments:

Reviewer's Responses to Questions

**Comments to the Author**

1. Does the manuscript provide a valid rationale for the proposed study, with clearly identified and justified research questions?

Reviewer #1: Yes

Reviewer #2: Yes

2. Is the protocol technically sound and planned in a manner that will lead to a meaningful outcome and allow testing the stated hypotheses?

Reviewer #1: Yes

Reviewer #2: Yes

3. Is the methodology feasible and described in sufficient detail to allow the work to be replicable?

Reviewer #1: Yes

Reviewer #2: Yes

4. Have the authors described where all data underlying the findings will be made available when the study is complete?

Reviewer #1: No

Reviewer #2: Yes

5. Is the manuscript presented in an intelligible fashion and written in standard English?

Reviewer #1: Yes

Reviewer #2: Yes

6. Review Comments to the Author

You may also provide optional suggestions and comments to authors that they might find helpful in planning their study.

Reviewer #1: Introduction

Well-structured and focused on the topic under study.

Methodology

Eligibility of research questions: Although Population is not applicable, the study element Policies and actions should be included in the inclusion criteria. Otherwise, it would be incomplete, as it would only indicate that it is included in the concept of climate change adaptation and focused on diseases of infectious origin.

The inclusion criteria that sources must meet to be included in the Scoping Review should also be indicated.

Search strategy

Using the keywords indicated, a basic search should be carried out in each database, allowing other keywords (synonyms and database-specific) to be identified.

The search strings to be used specifically in each database should be indicated.

Study selection and eligibility criteria

Specify the time period over which the search will be conducted.

Justify why papers from 2007 to 2022 will be selected.

Analysis

Will the articles be analysed by at least two reviewers? What degree of agreement between the reviewers will be necessary to include a study in the scoping review? This should be explained, given its importance for the rigour of the scoping review.

Explain how the PRISMA flowchart will be used.

Reviewer #2: My Comments:

Title : Because this scoping review only focus on infectious diseases control, may I suggest to add this focus in title.

Rational of the Objective is clear: systematic comprehensive scoping review of current policies and programs on the climate change adaptation in the ASEAN health systems, with particular focus on policies related to infectious diseases control using JBI methodology based on the WHO operational framework for building climate-resilient health systems.

Introduction is a good sequence, problem description, available knowledge, summary of what is currently known about the problem and rationale with specific aims.

Method: Clear and concise research question, sub questions and eligible criteria.

Regarding “eligibility of the research questions”, authors did mentioned for “population” was not applicable. But I think authors can specified “Human Population” because climate change can effect on both human and animal populations. For this scoping review, I believe authors just want to review on human population. Both concepts and contexts are clear.

Search database and searching strategies seem clear. Protocol will be more comprehensive if authors can show the detail keywords (with synonyms will be used). For example, in “infectious diseases” keywords, I think, authors will review only on climate-related infectious diseases, not including all infectious diseases happing in the ASEAN regions.

Authors planned to reviews from 2007 to 2022, which seems okay. I am just curious how authors will plan to handle the review if one ASEAN country was changed their policy between these periods. There might have 2 or 3 polices for one ASEAN country within these periods. Authors also plan to map the characteristics of exiting policies in each ASEAN country, if possible, added more variable that which police was developed by which Ministry.

Again, in one country, there might have several co-developed polices implemented by different Ministries. Authors were suggested to map all these co-developed polices focus on climate change and their inter-related issues.

Data extraction seems okay, expect characteristics of policies wasn’t collected, which is included in research sub-question. If one country has changed polices within 2007 to 2022, authors should collect and mention about this.

Both strength and limitation were addressed. I don’t know why authors mentioned about “English Keywords”. Because authors already mentioned in study selection that “No language restrictions will be imposed”.

Reporting Publication Guideline: It is very well that authors will use PRISMA-ScR for reporting of publication guideline. Right Tool was used. Great !

7. PLOS authors have the option to publish the peer review history of their article (what does this mean?). If published, this will include your full peer review and any attached files.

Reviewer #1: No

Reviewer #2: **Yes: **Dr. Win Khaing

---

## [Author Response · Author response to Decision Letter 0]

2 May 2023

Dear Editor,

Thank you for your email enclosing the reviewers’ comments for our manuscript “Adaptation of health systems to climate change-related infectious disease outbreaks in the ASEAN: Protocol for a scoping review of national and regional policies.”

We have carefully reviewed the comments and have revised the manuscript accordingly. Our responses are given in a point-by-point manner below.

We are pleased to submit our revised manuscript.

We hope the revised version is now suitable for publication, and we look forward to hearing from you in due course. The final draft has been approved by all the authors. The material has not and will not be offered elsewhere for possible publication as long as it is under your journal’s consideration. 

Sincerely,

Authors

JOURNAL REQUIREMENTS

Response: Thank you. We have modified the file naming and manuscript style to meet the PLOS ONE’s style requirement.

2. Please submit the filled PRISMA- ScR checklist

Response: Thank you. We have submitted the filled checklist.

3. Please upload a copy of your study protocol that was approved by your ethics committee/IRB as a Supporting Information file. By the study protocol, we mean the complete and detailed plan for the conduct and analysis of the trial approved by the ethics committee/IRB. Please send this in the original language. If this is in a language other than English, please also provide a translation. [https://journals.plos.org/plosone/s/submission-guidelines#loc-guidelines-for-specific-study-types]

Response: Thank you for this comment. As this is a scoping review protocol, ethics approval is not needed from our IRB.

Response: Thank you for this comment. We have addressed this comment.

REVIEWER #1: 

Have the authors described where all data underlying the findings will be made available when the study is complete? No

The PLOS Data policy requires authors to make all data underlying the findings described in their manuscript fully available without restriction, with rare exception, at the time of publication. The data should be provided as part of the manuscript or its supporting information, or deposited to a public repository. For example, in addition to summary statistics, the data points behind means, medians and variance measures should be available. If there are restrictions on publicly sharing data—e.g. participant privacy or use of data from a third party—those must be specified

Response: Thank you for your comment. We have added a data policy statement in the ‘Ethics and dissemination’ section (Line 173-175): “All relevant literature search results such as government documents, journal articles, and publications, will be made available in a public repository.”

Introduction

1. Well-structured and focused on the topic under study.

Response: Thank you for your appreciative comment.

Methodology

2. Eligibility of research questions: Although Population is not applicable, the study element Policies and actions should be included in the inclusion criteria. Otherwise, it would be incomplete, as it would only indicate that it is included in the concept of climate change adaptation and focused on diseases of infectious origin.

Response: Thank you for your comment. We have added the word ‘policies and actions’ in the Concept aspect of the PCC framework. (Line 118). We have also added the Population aspect in the PCC framework as per the suggestion of Reviewer #2. (Line 117)

3. The inclusion criteria that sources must meet to be included in the Scoping Review should also be indicated.

Response: Thank you for your comment. The inclusion and exclusion criteria have been previously defined in the section “Study selection and eligibility criteria” (Line 134-146). We have rephrased the section for better clarity. 

Search strategy

4. Using the keywords indicated, a basic search should be carried out in each database, allowing other keywords (synonyms and database-specific) to be identified.

The search strings to be used specifically in each database should be indicated.

Response: Thank you for your comment. We have conducted a basic search in each database and included the search keywords that will be used in each database in S1 Appendix. Additionally, we have clarified the name of the database ‘WHO Library’ to be the ‘World Health Organization (WHO) Institutional Repository Information Sharing (IRIS)’.

5. Study selection and eligibility criteria

Specify the time period over which the search will be conducted.

Response: Thank you for your comment. We have included the exact period of the included studies in Line 138-139 (between January 2003-January 2023).

6. Justify why papers from 2007 to 2022 will be selected.

Response: Thank you for your comment. After further deliberation, we believe it is better to include papers that were published in the last 20 years (2003-2023). With this time duration, we aim to comprehensively capture the latest and current climate change adaptation policies in ASEAN, in particular in the field of infectious diseases. (Line 138-140)

Analysis

7. Will the articles be analysed by at least two reviewers? What degree of agreement between the reviewers will be necessary to include a study in the scoping review? This should be explained, given its importance for the rigour of the scoping review.

Response: Thank you for your comment. We have added this information in the protocol. “Two reviewers, AVM and BWL, will independently assess the eligibility and inclusion of the studies by screening the titles and abstracts, followed by full-text assessments. Any differences will be resolved through consensus.” (Line 144-146)

8. Explain how the PRISMA flowchart will be used.

Response” Thank you for your comment. We have added an explanation in Line : “To summarize our search, we will use the PRISMA flow chart.” (Line 149)

REVIEWER #2

1. Title : Because this scoping review only focus on infectious diseases control, may I suggest to add this focus in title.

Response: Thank you for your comment. We have changed the title to “Adaptation of health systems to climate change-related infectious disease outbreaks in the ASEAN: Protocol for a scoping review of national and regional policies”. (Line 4-6)

2. Rational of the Objective is clear: systematic comprehensive scoping review of current policies and programs on the climate change adaptation in the ASEAN health systems, with particular focus on policies related to infectious diseases control using JBI methodology based on the WHO operational framework for building climate-resilient health systems.

Response: Thank you for your appreciative comment.

3. Introduction is a good sequence, problem description, available knowledge, summary of what is currently known about the problem and rationale with specific aims.

Response: Thank you for your appreciative comment.

3. Method: Clear and concise research question, sub questions and eligible criteria.

Response: Thank you for your appreciative comment.

4. Regarding “eligibility of the research questions”, authors did mentioned for “population” was not applicable. But I think authors can specified “Human Population” because climate change can effect on both human and animal populations. For this scoping review, I believe authors just want to review on human population. Both concepts and contexts are clear.

Response: Thank you for your comment. We have added ‘humans’ in the PCC framework. (Line 117)

5. Search database and searching strategies seem clear. Protocol will be more comprehensive if authors can show the detail keywords (with synonyms will be used). For example, in “infectious diseases” keywords, I think, authors will review only on climate-related infectious diseases, not including all infectious diseases happing in the ASEAN regions.

Response: Thank you for your comment. We have now included this in S1 Appendix. We hope that by including the Boolean ‘AND’ (“climate change AND infectious diseases”), only climate-related infectious diseases will be included in the search results.

6. Authors planned to reviews from 2007 to 2022, which seems okay. I am just curious how authors will plan to handle the review if one ASEAN country was changed their policy between these periods. There might have 2 or 3 polices for one ASEAN country within these periods.

Response: Thank you for your comment. After further deliberation, we have changed the duration to 20 years to allow for more comprehensive analysis of the current policy landscape in the ASEAN. We have also included an explanation on how we will handle policy change: “If there are two or more policies published by a country or in ASEAN, we will only include policies that are still in effect.” (Line 140-141)

7. Authors also plan to map the characteristics of exiting policies in each ASEAN country, if possible, added more variable that which police was developed by which Ministry.

Again, in one country, there might have several co-developed polices implemented by different Ministries. Authors were suggested to map all these co-developed polices focus on climate change and their inter-related issues.

Response: Thank you for your comment.We have added a category in the extracted data called “Issuing body” (Line 151, with a detailed description provided in Table 1).

8. Data extraction seems okay, expect characteristics of policies wasn’t collected, which is included in research sub-question. If one country has changed polices within 2007 to 2022, authors should collect and mention about this.

Response: Thank you for your comment. We aim the collect the characteristics of the current policies, namely those included in Table 1 (type of document, title of document, issuing body, and type of infectious diseases addressed). Now, we have also included the policy change information as the data that we want to collect for the review (Line 156-157)

9. Both strength and limitation were addressed. I don’t know why authors mentioned about “English Keywords”. Because authors already mentioned in study selection that “No language restrictions will be imposed”.

Response: Thank you for your comment. We include this because we plan to only use English keywords for the search, and not keywords in other languages. This may cause the search results not to include all available policies, as those without English titles or English keywords in the documents may not be included. We have further clarified this in Line 185.

10. Reporting Publication Guideline: It is very well that authors will use PRISMA-ScR for reporting of publication guideline. Right Tool was used. Great !

Response: Thank you for your appreciative comment.

---

## [Decision Letter · Decision Letter 1]

25 May 2023

Adaptation of health systems to climate change-related infectious disease outbreaks in the ASEAN: Protocol for a scoping review of national and regional policies

PONE-D-23-02161R1

Dear Dr. Miranda,

We’re pleased to inform you that your manuscript has been judged scientifically suitable for publication and will be formally accepted for publication once it meets all outstanding technical requirements.

Kind regards,

Seo Ah Hong, PhD

Academic Editor

PLOS ONE

Additional Editor Comments (optional):

Reviewers' comments:

Reviewer's Responses to Questions

**Comments to the Author**

1. Does the manuscript provide a valid rationale for the proposed study, with clearly identified and justified research questions?

Reviewer #1: Yes

Reviewer #2: Yes

2. Is the protocol technically sound and planned in a manner that will lead to a meaningful outcome and allow testing the stated hypotheses?

Reviewer #1: Yes

Reviewer #2: Yes

3. Is the methodology feasible and described in sufficient detail to allow the work to be replicable?

Reviewer #1: Yes

Reviewer #2: Yes

4. Have the authors described where all data underlying the findings will be made available when the study is complete?

Reviewer #1: Yes

Reviewer #2: Yes

5. Is the manuscript presented in an intelligible fashion and written in standard English?

Reviewer #1: Yes

Reviewer #2: Yes

6. Review Comments to the Author

You may also provide optional suggestions and comments to authors that they might find helpful in planning their study.

Reviewer #1: Following the second revision, the authors have responded adequately to my comments and considerations and have substantially improved the manuscript.

Reviewer #2: All my comments have been addressed and the necessary modifications have been made. Nothing more comment. Good job!

7. PLOS authors have the option to publish the peer review history of their article (what does this mean?). If published, this will include your full peer review and any attached files.

Reviewer #1: No

Reviewer #2: **Yes: **Win Khaing

---

## [Editor Report · Acceptance letter]

29 May 2023

PONE-D-23-02161R1 

Adaptation of health systems to climate change-related infectious disease outbreaks in the ASEAN: Protocol for a scoping review of national and regional policies 

Dear Dr. Miranda:

I'm pleased to inform you that your manuscript has been deemed suitable for publication in PLOS ONE. Congratulations! Your manuscript is now with our production department. 

Kind regards, 

on behalf of

Prof. Seo Ah Hong 

Academic Editor

PLOS ONE